# Prognostic Hematologic Biomarkers Following Immune Checkpoint Inhibition in Metastatic Uveal Melanoma

**DOI:** 10.3390/cancers14235789

**Published:** 2022-11-24

**Authors:** Jessica J. Waninger, Leslie A. Fecher, Christopher Lao, Sarah Yentz, Michael D. Green, Hakan Demirci

**Affiliations:** 1Medical Scientist Training Program, University of Michigan, Ann Arbor, MI 48109, USA; 2Department of Internal Medicine, Division of Hematology Oncology, Rogel Cancer Center, Ann Arbor, MI 48109, USA; 3Department of Radiation Oncology, University of Michigan, Ann Arbor, MI 48109, USA; 4Department of Microbiology and Immunology, University of Michigan, Ann Arbor, MI 48109, USA; 5Department of Radiation Oncology, Veterans Affairs Ann Arbor Healthcare System, Ann Arbor, MI 48105, USA; 6Department of Ophthalmology and Visual Sciences, University of Michigan, Ann Arbor, MI 48105, USA

**Keywords:** ocular melanoma, uveal melanoma, NLR, LDH, prognosis, immune checkpoint inhibitors

## Abstract

**Simple Summary:**

Uveal melanoma is a rare form of melanoma but is the most common tumor in the eye. Despite having effective treatments for the initial tumor, many patients experience the spread of their cancer to distant body sites. There is no uniform way of treating metastatic disease, but physicians often use therapies that harness the patient’s immune system because these same treatments have been very effective in other types of melanoma. Not all patients respond to this therapy though, and some develop toxicity related to the treatment. The goal of this paper was to identify features or blood markers that may help determine response to treatment early. Specifically, we analyzed a molecule called lactate dehydrogenase (LDH) and the ratio of different white blood cells at the start of therapy and 2 months after treatment was started. We found that these non-invasive blood markers could be useful in determining which patients are responding to treatment.

**Abstract:**

Background: There is no standardized treatment for metastatic uveal melanoma (MUM) but immune checkpoint inhibitors (ICI) are increasingly used. While ICI has transformed the survival of metastatic cutaneous melanoma, MUM patients do not equally benefit. Factors known to affect ICI response include the hematologic markers, lactate dehydrogenase (LDH) and neutrophil:lymphocyte ratio (NLR). We evaluated the prognostic value of LDH and NLR at the start of ICI and on treatment in MUM. Methods: MUM patients were treated between August 2006 and May 2022 with combination ipilimumab/nivolumab or ipilimumab/nivolumab/pembrolizumab single-agent therapy. Univariable (UVA) and multivariable (MVA) analyses were used to assess the prognostic value of predefined baseline factors on progression-free (PFS) and overall survival (OS). Results: In forty-six patients with MUM treated with ICI, elevated baseline and on-treatment LDH was prognostic for OS (start of ICI, HR (95% CI): 3.6 (1.9–7.0), *p* < 0.01; on-treatment, HR (95% CI): 3.7 (1.6–8.8), *p* < 0.01) and PFS (start of ICI, (HR (95% CI): 2.8 (1.5–5.4), *p* < 0.0001); on-treatment LDH (HR (95% CI): 2.2 (1.1–4.3), *p* < 0.01). On-treatment NLR was prognostic for PFS (HR (95% CI): 1.9 (1.0–3.9), *p* < 0.01). On-treatment LDH remained an important contributor to survival on MVA (OS: HR (95% CI): 1.001 (1.00–1.002), *p* < 0.05); PFS: HR (95% CI): 1.001 (1.00–1.002), *p* < 0.01). Conclusions: This study demonstrates that LDH and NLR could be useful in the prognostication of MUM patients treated with ICI. Additional studies are needed to confirm the importance of these and other prognostic biomarkers.

## 1. Introduction

Uveal melanoma (UM) is the most common intraocular malignancy [1]. The current primary tumor treatments—plaque radiotherapy and enucleation—provide 95–99% local tumor control. Despite good local control, systemic prognosis remains poor, as nearly half of patients die from metastatic disease within 10 years of their original diagnosis [2,3,4]

To date there is no standardized treatment algorithm for metastatic UM (MUM). Approximately 90% of UMs harbor driver mutations in GNAQ/GNA11, which are genes that code for Gα proteins that mediate multiple downstream signaling cascades including the mitogen activated kinase pathway (MAPK). The MAPK pathway and its components have been the focus in the development of targeted therapies in cutaneous melanoma but when applied to patients with MUM, no survival benefit has yet been shown [5,6]. For example, when compared to chemotherapy, selumetinib, a MAP/ERK kinase (MEK) inhibitor, did not show a statistically significant improvement in overall survival (OS) (11.8 vs. 9.1 months) [7]. This drug has also been studied in combination with the cytotoxic agent, dacarbazine, in a phase III randomized controlled trial. Similarly, no significant difference in progression-free survival (PFS) or objective response rate (3 vs. 0%) was seen [8].

Based on their effectiveness in advanced stage cutaneous melanoma, immune checkpoint inhibitors (ICI), such as those blocking anti-cytotoxic T-lymphocyte-associated protein 4 (anti-CTLA-4) and/or programmed cell death-1 (PD-1)/programmed cell death ligand-1 PD-L1) are widely used in the treatment of MUM [9,10,11]. While these agents have revolutionized patient outcomes for advanced stage cutaneous melanoma as well as improved outcomes in mucosal melanoma, patients with MUM do not derive equal benefit. One study prospectively evaluated the use of pembrolizumab as a first-line therapy. Overall survival (OS) for patients who derived objective clinical benefit was 12.8 months, which is similar to other agents [12]. A small phase II study evaluating combination ipilimumab 3 mg/kg and nivolumab 1 mg/kg reported an overall response rate (ORR) of 18% including one complete response (CR) and a median OS of 19.1 months [13]. Forty percent of patients in this trial experienced a grade 3–4 treatment-related adverse event, which is expected for this treatment regimen and is consistent with other reports [14,15,16]. These high toxicity rates highlight the importance of identifying patients who will benefit from ICI.

Features thought to positively influence response to ICI include high tumor mutational burden, high expression of PD-1, low lactate dehydrogenase (LDH), and the absence of liver metastases (LM)—features not typically present in uveal melanoma patients [17,18,19,20,21,22,23,24,25]. In fact, the first and most common site of metastasis for patients with UM is the liver with >90% of patients developing tumors in this anatomic location [26]. Liver-directed therapies are commonly used in the treatment of MUM patients with limited LM. For patients with resectable disease, liver resection has shown a survival benefit. One retrospective study reported a post operative median OS of 14 months for all patients; if complete resection was possible the median OS increased to 27 months [27]. Other regional liver directed therapies like transarterial chemoembolization (TACE), hepatic artery infusion (HAI), or selective internal radiotherapy result in a broad range of survival outcomes [28]. A separate single institution retrospective study reported that local therapy (surgery or intrahepatic chemotherapy) correlated with prolonged survival on univariate and multivariate analyses (median OS 32.4 months) [29].

Another feature thought to be prognostic and predictive of ICI response across a variety of tumor types, including metastatic melanoma, is an elevated ratio of neutrophils to lymphocytes (NLR) at the start of therapy [30,31,32,33]. The change in NLR in response to ICI has also been linked to early response to therapy [34,35]. Despite its proven utility in other cancer types, there is only one published study that assessed NLR in patients with metastatic uveal melanoma and, importantly, this was not in the context of immunotherapy [36]. While there has been some effort to understand and identify features that impact survival in MUM, there are few studies that assess these factors in patients receiving ICI [37]. Furthermore, even fewer or these studies were conducted in the first-line setting, as most evaluated ICI agents as a salvage therapy.

To gain insight into factors that may be prognostic for response to ICI in MUM we conducted a real-world review and retrospective analysis of all MUM patients who were treated with either single-agent or combination ICI. We first evaluated the established patient demographics, primary tumor features, and treatment characteristics in the prognostication of MUM patients and extended this analysis to the hematologic biomarkers, LDH and NLR, at baseline and again while on treatment. We determined the objective tumor response for each patient and examined distinguishing characteristics of patients who benefited from ICI. Furthermore, we describe our real-world institutional experience treating patients with this complicated disease.

## 2. Materials and Methods

### 2.1. Patient Population and Data Sources

We performed a single-center, retrospective analysis of 46 metastatic uveal melanoma patients who received immunotherapy between September 2012 and May 2022. All these patients were treated with either single agent ipilimumab, nivolumab, or pembrolizumab therapy or ipilimumab/nivolumab combination therapy. Patients and data were collected via the University of Michigan electronic medical record (EMR) system. Patient data was extracted manually. All clinical records were obtained with the approval of Institutional Review Boards and patients’ consents were waived following Institutional Review Board protocol review (HUM00163915, HUM00046408).

### 2.2. Data Collection and Treatment Outcomes

Patient demographic information and tumor characteristics included age, sex, Eastern Cooperative Oncology Group (ECOG) performance status, ocular location of the primary tumor (iris, ciliary body, choroid), ciliary body involvement, extraocular extension, tumor thickness, longest basal diameter, gene expression profile (GEP) class, PRAME status, histopathology, time from primary diagnosis to metastatic relapse and metastatic disease sites at the start of ICI (baseline). Hematologic markers collected included LDH at the following time points: metastatic disease diagnosis, baseline, and 8 weeks post ICI-start (on treatment), as well as absolute lymphocyte count (ALC), absolute neutrophil count (ANC), and absolute eosinophil count (AEC) at baseline and on treatment. A hard cut-off of 60 days following therapy start was used for all “on treatment” assessments. Patients who did not have this time-point were excluded from the analysis. NLR was defined as the ratio of the ANC to the ALC in each peripheral blood sample. dNLR was calculated using a formula previously demonstrated [32]:dNLR = ANC/(white blood cell count – ANC). Delta NLR was calculated as the difference between on treatment and baseline NLR. Treatment characteristics collected included treatment of the primary (enucleation vs. plaque RT), liver-directed therapy type, lines of prior therapy, immunotherapeutic agent(s), cycles of ICI completed, reason for ICI discontinuation, immune related adverse event (IRAE) grade, and objective response using the RECIST criteria (version 1.1) [38]. For the purposes of this study, responders were designated as those who derived clinical benefit from therapy, i.e., those who had complete response (CR), partial response (PR), or stable disease (SD). Non-responders were patients who had progressive disease (PD).

The endpoints analyzed included OS or PFS. Date of radiographic progression was determined by manual review of radiological reports and date of death was determined by manual review of the EMR. Survival time was measured from the start of immunotherapy to the date of death (for OS) or to disease progression (for PFS). Death certificates and hospital encounters at the end of life were reviewed. All patients had either (1) progressive disease radiographically, (2) progressive symptoms felt related to their cancer, or (3) cancer listed as a primary or secondary cause of death. Patients were censored at the date of last known follow-up, defined as the most recent encounter with a documented provider. Patients with missing values pertinent to the specific analysis were excluded. Two of 46 patients (4.3%) were lost to follow up. 

### 2.3. Statistical Analysis

OS and PFS estimates were generated with Kaplan–Meier method. Comparison of survival outcomes between groups and hazard ratios (HR) were generated using Gehan-Breslow-Wilcoxon and Mantel-Haenszel tests in univariable analyses. LDH was split into two groups based on the “normal” cutoff used at the University of Michigan: below 240 mg/dL and greater than or equal to 240 mg/dL. NLR was also split into two groups at the median NLR value for this cohort at each specified time-point. Multivariable Cox regression was performed to estimate the effect of each measure on survival. Continuous variables included longest basal diameter, tumor thickness, baseline and on treatment NLR as well as LDH at metastatic diagnosis, baseline, and while on treatment. The categorical variables, concurrent ipilimumab/nivolumab (vs. single agent ICB) and ECOG performance status (0 vs. >0), were also included in the model. HRs and 95% confidence intervals (CI) for each measure were generated for the overall cohort as well as subgroup analyses from the interaction term by the Wald method. For investigation of possible differences in baseline characteristics between responders (designated as CR, PR, SD) versus non-responders (designated as PD), Fischer-exact test was used for categorical variables and independent *t*-test for continuous variables. In all cases, two-tailed *p*-values were calculated with a significance cut-off of *p* < 0.05. All analyses were conducted using SPSS statistical software (IBM Corp, version 28.0.1.0, Armonk, NY, USA) and images were created in GraphPad Prism (GraphPad Software, version 8.0.0, San Diego, CA, USA) and Adobe Illustrator (version 26.4.1, San Jose, CA, USA).

## 3. Results

### 3.1. Patient, Tumor, and Treatment Characteristics

A total of 46 patients with biopsy-confirmed metastatic UM were included in this analysis (Table 1, Table 2 and Table 3, Appendix A). The median age was 61.8 years (IQR = 20.0). Twenty-six patients (56.5%) were male and 20 (43.5%) were female. At the time of diagnosis of primary UM, 43 (91.5%) melanomas originated from the choroid and 21 (45.7%) had ciliary body involvement. Three (6.5%) patients had extraocular extension. Twenty-six (56.5%) patients received plaque radiotherapy and 24 (52.2%) were enucleated. Of those patients that underwent enucleation, 11 patients had mixed cell, 4 spindle cell, and 8 had epithelioid cell type UM.

Given that many of these patients were diagnosed prior to routine genetic data collection, only 32 (69.6%) had gene expression profiling (GEP) and 10 (21.7%) had a PRAME status. Two (4.3%) patients were class Ia, 2 (4.3%) were class Ib, and 28 (60.9%) were class II. In this cohort, a median of 31 months (IQR = 51) elapsed between the diagnosis of their primary tumor and the identification of metastatic disease. At the time their metastatic disease was diagnosed, most patients had liver metastases (LM, *n* = 41, 89.1%) and 14 (30.4%) of these patients did not have any evidence of extra-hepatic disease. At various points during their treatment courses, 23 (50.0%) patients were treated with a liver-directed therapy: surgical resection (2, 4.3%), SBRT/RT (9, 19.6%), chemoembolization (6, 13.0%), radioembolization (1, 2.2%), or a combination of these therapies (5, 10.9%). Regarding systemic therapy, 43 (93.5%) patients received ICI as a first-line therapy. Combination ipilimumab/nivolumab was administered to 11 (23.9%) patients and the remaining received initial single agent therapy (35, 76.1%). Thirty-one (67.4%) patients had progression of their MUM while on therapy while 11 (23.9%) experienced treatment-related toxicities that required treatment discontinuation. 

### 3.2. Survival Outcomes

The median OS for the entire cohort was 11.4 months (95% CI: 7.5–21.5 months) with a 49% survival rate at one year. Most patients had progressed by 6 months after the start of ICI (median PFS: 3.2 months, 95% CI: 2.6–5.1 months) (Figure 1). To identify factors that may be prognostic for survival or response to ICI we completed univariable analyses on patient demographic factors, primary tumor characteristics, treatment characteristics and hematologic lab values at baseline and while on treatment. Of the patient demographic and primary tumor features, the only factor that significantly impacted OS was primary tumor thickness, HR (95% CI): 0.9 (0.8–1.0), *p* = 0.033. Primary tumor thickness was also prognostically significant for PFS: HR (95% CI): 0.9 (0.8–1.0), *p* = 0.038. Among extrahepatic metastatic sites, patients with lymph node metastases had worse OS, HR (95% CI): 2.1 (1.1–4.3), *p* = 0.031. Analysis of other tumor features including ciliary body involvement, longest basal diameter, and presence of liver metastases did not reach significance our cohort (Table 1, Table 2 and Table 3, univariate analyses). For patients that did have a GEP class, there was no statistically significant difference between risk stratification groups and survival outcomes [1a vs. 2: OS, HR (95% CI): 3.5 (0.5–25.9), *p* = 0.87; PFS, HR (95% CI): 3.1 (0.7–13.0), *p* = 0.64); 1b vs. 2: OS, HR (95% CI): 1.1 (0.2–5.1), *p* = 0.59; PFS, HR (95% CI): 1.2 (0.3–4.7), *p* = 0.76); 2 vs. 1 (all): OS, HR (95% CI): 1.3 (0.4–3.9), *p* = 0.63; PFS, HR (95% CI): 0.9 (0.3–2.6), *p* = 0.95)] (Table 1).

### 3.3. Hematologic Markers Prognostic for ICI-Response

Hematologic biomarkers including LDH and NLR have been shown to be both prognostic and predictive of ICI response in other tumor types. We sought to evaluate if these markers had any prognostic value in MUM. LDH at the time of metastatic disease diagnosis (HR (95% CI): 2.2 (1.1–4.3), *p* = 0.038), at baseline (HR (95% CI): 3.6 (1.9–7.0), *p* = 0.0011), and on treatment (HR (95% CI): 3.7 (1.6–8.8), *p* = 0.0046) were all prognostically significant for OS (Figure 2A,B). Of the three time-points analyzed, only baseline (HR (95% CI): 2.8 (1.5–5.4), *p* < 0.0001) and on treatment LDH (HR (95% CI): 2.2 (1.1–4.3), *p* = 0.0014) were prognostic for PFS (Figure 2C,D). NLR was also analyzed at baseline and on treatment. Baseline NLR did not appear to be prognostic for OS (HR (95% CI): 1.0 (0.5–2.0), *p* = 0.98) or PFS (HR (95% CI): 1.3 (0.7–2.5), *p* = 0.18) (Figure 2E,G). When evaluating on treatment values, NLR did not reach significance for OS (HR (95% CI): 1.6 (0.8–3.4), *p* = 0.057) but was prognostic for PFS (HR (95% CI): 1.9 (1.0–3.9), *p* = 0.0098) (Table 3, Figure 2F,H).

On multivariable analysis, only on treatment LDH values significantly contributed to survival outcomes [OS: (HR (95% CI): 1.001 (1.00–1.002), *p* = 0.017); PFS: (HR (95% CI): 1.001 (1.00–1.002), *p* = 0.004)] (Figure 3A,B). Another important prognostic factor for PFS was ECOG performance status (HR (95% CI): 5.26 (1.18–25.5), *p* = 0.030 (Figure 3B).

### 3.4. Patient Factors Associated with Clinical Benefit from ICI

To objectively evaluate treatment response in this cohort, we used RECIST criteria to quantitatively measure each patient’s tumor at the start of immunotherapy and again at their next imaging assessment (9–12 weeks after therapy initiation). Of the 46 patients included in this study, 2 did not have available imaging and 5 had clinical and/or radiographic progression that was not quantifiable. In total, 34 (73.9%) patients had progressive disease (PD) while 10 (21.7%) patients derived a clinical benefit from therapy (i.e., CR, PR, or SD). There was one patient who achieved a CR (2.2%), one with PR (2.2%), and eight with SD (17.4%). The average time to progression for those with SD was 24.5 months after treatment initiation (IQR: 42 months) (Figure 4A).

We sought to better understand if any specific patient, tumor, treatment or hematologic factors were related to clinical benefit from treatment. A greater proportion of patients in the responders group received combination ipilimumab/nivolumab as their first-line agent (40% vs. 20%, *p* = 0.24) (Figure 4B, Appendix A, Appendix A, Appendix A). Moreover, patients who received combination or sequential anti-CTLA-4 and anti-PD-1 therapy at any point in their treatment course had significantly better OS (HR (95% CI): 2.3 (1.1–4.5), *p* = 0.0012). Median survival was 23.6 months in this group compared to 5.6 months for patients who only received monotherapy. Median PFS for patients who received combination therapy was 3.9 months compared to 2.7 months, however, this difference did not reach significance (HR (95% CI): 1.7 (0.9–3.1), *p* = 0.09) (Appendix A).

Of the demographic variables analyzed there were more people with poor performance status in the non-responder (*p* = 0.046). Primary tumor characteristics including tumor thickness and longest basal diameter were not different between the groups (*p* = 0.78 and *p* = 0.75, respectively). Patients who benefited from therapy had a longer latent period between the diagnosis of their primary lesion and the development of metastases (64 vs. 22 months, *p* = 0.32) and fewer patients (70% vs. 94.1%, *p* = 0.069) had liver metastases at the start of immunotherapy. Responders also had lower median LDH at stage IV diagnosis (178 vs. 211 mg/dL, *p* = 0.14)) and while on treatment (202 vs. 258 mg/dL, *p* = 0.24) however, none of these reached statistical significance. Similarly, while median on treatment NLR values were higher for the non-responder group, the mean difference was not significant (Baseline, 2.6 vs. 2.6, *p* = 0.85; On treatment, 2.0 vs. 3.3, *p* = 0.47).

## 4. Discussion

In this study of 46 metastatic uveal melanoma patients, the median OS was 11.4 months. We describe factors, including the hematologic markers LDH and NLR, that may be important for prognostication in patients who receive ICI. We also describe differences in clinicopathologic and treatment characteristics between patients who had an objective benefit from immunotherapy and those who did not. 

LDH has long been shown to be an important prognostic marker in cancer due to its facilitation of glycolysis [18,39,40] and has also been show to modify the immune microenvironment, potentially impacting the efficacy of ICI [41,42]. Its prognostic value in cutaneous melanoma is well established and included in the AJCC staging system [43]. Recently published studies of MUM patients, including a phase 2 clinical trial evaluating the efficacy of pembrolizumab, also demonstrate LDH to be prognostic [14,44]. In our study we confirmed the prognostic importance of LDH in MUM at various points in the clinical disease course. Furthermore, patients who benefited from ICI had lower median on treatment LDH values.

Another prognostically valuable hematologic marker that has been identified for solid tumors, including metastatic cutaneous melanoma, is NLR [31,32]. While NLR has been shown to be prognostically important in other contexts, such as COVID-19 infection [45], recent studies showed independent value in cancer patients with NSCLC treated with adjuvant immunotherapy [34]. In our review of the literature very few studies have evaluated the role of NLR in UM patients. A recent study by Meijer et. al, examined the association between NLR and other systemic inflammatory markers, including erythrocyte sedimentation rate (ESR) and c-reactive protein (CRP), and metastasis-free survival at the time patients were treated for their primary tumor. While NLR was not a prognostic marker in this study, the authors showed that high CRP levels were associated with a longer metastasis-free survival [46]. To our knowledge, there is only one published report evaluating NLR in patients with MUM. This study examined NLR as it related to first-line treatment response, time-to-relapse after receiving first-line therapy, and OS [36]. NLR was not prognostic for treatment response but was significant in time-to-relapse and OS. Importantly, only 6.7% of patients included in this study received ICI, making it difficult to extrapolate these conclusions to patients receiving immunotherapy [36]. Our study demonstrated that NLR is a significant prognostic factor for PFS, particularly while patients are on ICI treatment, and therefore may be a useful aid in clinical decision making and patient stratification. Other hematologic markers including absolute cell counts, dNLR, and on treatment delta NLR were not significant in our study. While also not reaching significance in our split analysis, the median NLR was consistently lower in responders than in non-responders at both time points.

Liver metastases, which are the most common metastatic site in UM, have also been shown to impact response to ICI via systemic loss of antigen-specific T cells and possibly by other unknown mechanisms [22,47]. While not prognostic in our study, a greater percentage of non-responders had liver involvement. A separate study in MUM patients receiving ICI showed that patients who lived longer had extrahepatic metastases in addition to liver metastases [48]. While this finding may be due to biased selection for patients with more indolent disease, it is also possible that extrahepatic metastases facilitate tumor recognition by circulating immune cells, bypassing the immunosuppressive environment of the liver. This study also showed that patients who received liver-directed therapy had longer survival [48]. This finding was not reproduced in our patient cohort, likely due to a limited sample size. Currently, there is a phase I clinical trial investigating the feasibility of hepatic ablation of melanoma metastases, in conjunction with ipilimumab and nivolumab, to enhance immunotherapy efficacy (HAMMER trial; NCT05169957) [49]. Besides an increased proportion of patients with LM, UM has a significantly lower tumor mutational burden and less PD-1/PD-L1 expression when compared to cutaneous melanoma which may limit response to ICI [50,51].

Our institutional experience in treating MUM patients with ICI is consistent with other published reports at an overall one-year survival rate of approximately 49%. As the treatment strategy for metastatic melanoma has evolved over time, our approach to agent selection has also changed. In our cohort, a total of 9 patients were treated with initial anti-CTLA-4 therapy; 7 of which were treated prior to 2015. All but one of these patients progressed on therapy and were eventually switched to a PD-1 inhibitor (either pembrolizumab or nivolumab). Twenty-four of our patients received single-agent pembrolizumab as their first-line therapy, with mixed response. Consistent with other reports, patients who received combination or sequential anti-CTLA-4 and anti-PD-I therapy at any point in their treatment course tended to live longer with a median OS of 23.6 months vs. 5.6 months for patients who received life-time single-agent therapy (Appendix A) [14,52,53,54]. Though biased towards patients with indolent enough disease to receive additional therapies, these findings suggest that dual-agent therapy may be beneficial in this patient population.

### Limitations

The major limitations of this study are its small sample size, retrospective design, and lack of control or validation cohorts. When compared to other prospective and retrospective studies, the overall survival is similar at approximately one year. While the overall sample size was limited due to the rarity of the disease, the vast majority of patients included in this study received ICI as a front-line agent (93.6%), which is an improvement over many other published reports. Because GEP was not commercially available until late 2009, many of the patients included in this study did not have any available genetic data, including GEP class or PRAME status which together have shown prognostic value [55]. Recently, mutation of the methyl-binding domain 4 (MDB4) gene was identified as a predictive marker for immunotherapy response [56]. Encoding a glycosylase integral to DNA repair, it is suggested that defective MBD4 leads to increased tumor mutational burden, which in turn enhances the efficacy of immune checkpoint inhibitors. MDB4 mutational status, therefore, is an important confounder that we were unable to control for in this study. The prognostic and predictive value of CRP as a single pre-treatment measurement [57,58] and as a longitudinal metric [59] during ICI has been demonstrated for other tumor types. It would be interesting to evaluate CRP before and during treatment in metastatic uveal melanoma, however, our patient cohort did not have CRP consistently measured precluding us from inclusion of this important metric in our study. Furthermore, other potential prognostic biomarkers, tumor mutational burden, tumoral PD-L1, and HLA alleles were not available.

## 5. Conclusions

The findings of our study suggest that hematologic markers, such as LDH and NLR, may be helpful for patient prognostication and clinical decision making in patients with MUM being treated with immunotherapy. These agents are increasingly used in the treatment of MUM despite only modest improvement in survival outcomes. In this real-world analysis, we also demonstrated that patients who received initial combination ipilimumab/nivolumab tended to have better OS compared with patients who received initial monotherapy suggesting that combination therapy may be beneficial in the treatment of this deadly disease. Because immunomodulatory agents are not benign with respect to toxicity, more studies are needed to identify additional prognostic and predictive biomarkers that can aid in the identification of patients who will benefit from these therapies.

## Figures and Tables

**Figure 1 cancers-14-05789-f001:**
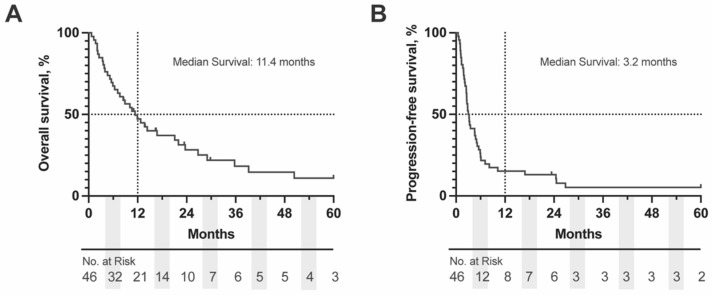
Kaplan-Meier (KM) Curve of (**A**) OS and (**B**) PFS for complete OM cohort. Dashed lines represent the 50% survival and 12 months, respectively. Number at risk is displayed below each graph in 6 months intervals. OS = overall survival, PFS = progression free survival.

**Figure 2 cancers-14-05789-f002:**
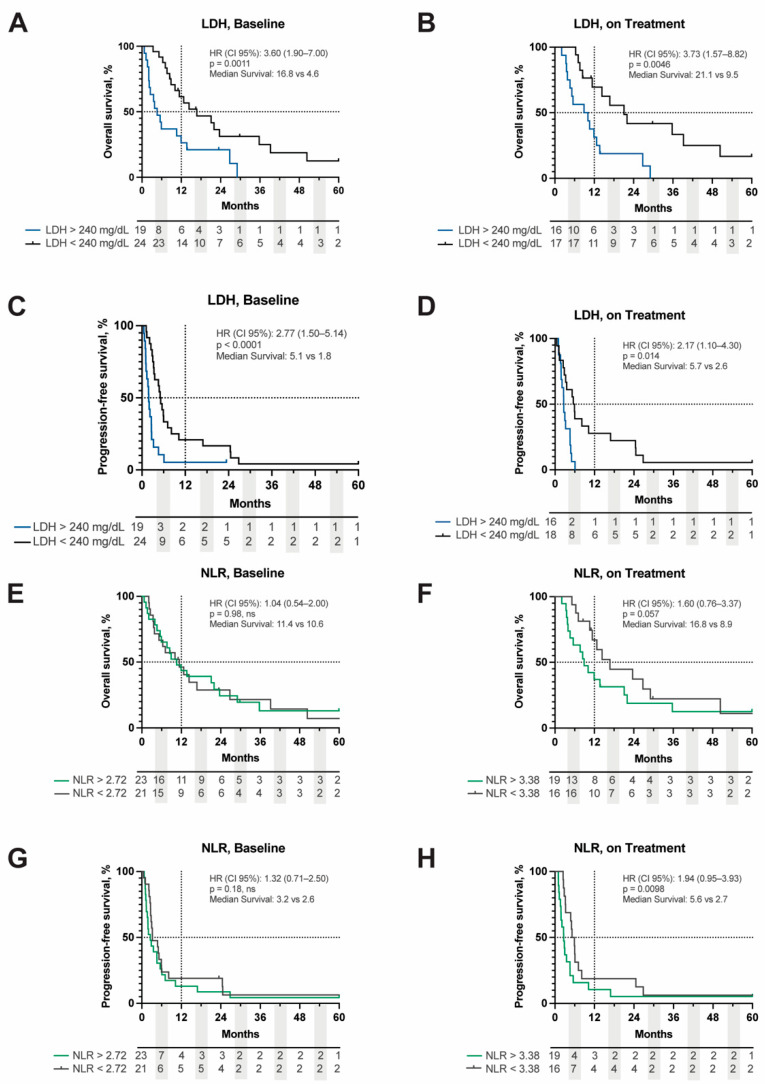
Kaplan–Meier curve of OS and PFS stratified by LDH value above or below 240 mg/dL at baseline (**A**,**C**) and on treatment (**B**,**D**). OS and PFS of NLR stratified by the median cut point of 2.72 at baseline (**E**,**G**) and 3.38 on treatment (**F**,**H**). Dashed lines represent 50%- and 12-month survival points, respectively. Hazard ratios (HR), *p*-values, and median survival are displayed in the upper right corner of each respective graph. Baseline = ICB start; on treatment = response assessment 8 weeks following ICB initiation. LDH = lactate dehydrogenase, NLR = neutrophil:lymphocyte ratio, OS = overall survival, PFS = progression-free survival.

**Figure 3 cancers-14-05789-f003:**
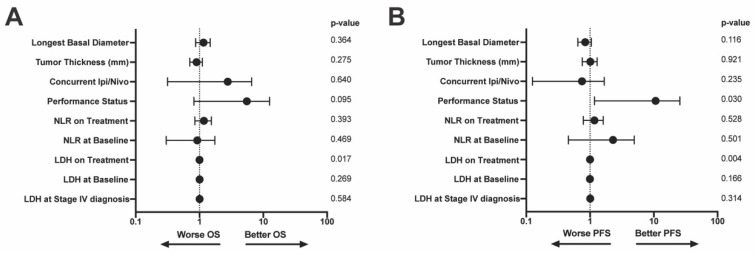
Multivariable model of (**A**) OS and (**B**) PFS. Variables included in the model are along the left y-axis. *p*-values associated with each variable are on the right y-axis. LDH = lactate dehydrogenase, NLR = neutrophil:lymphocyte ratio, OS = overall survival, PFS = progression free survival.

**Figure 4 cancers-14-05789-f004:**
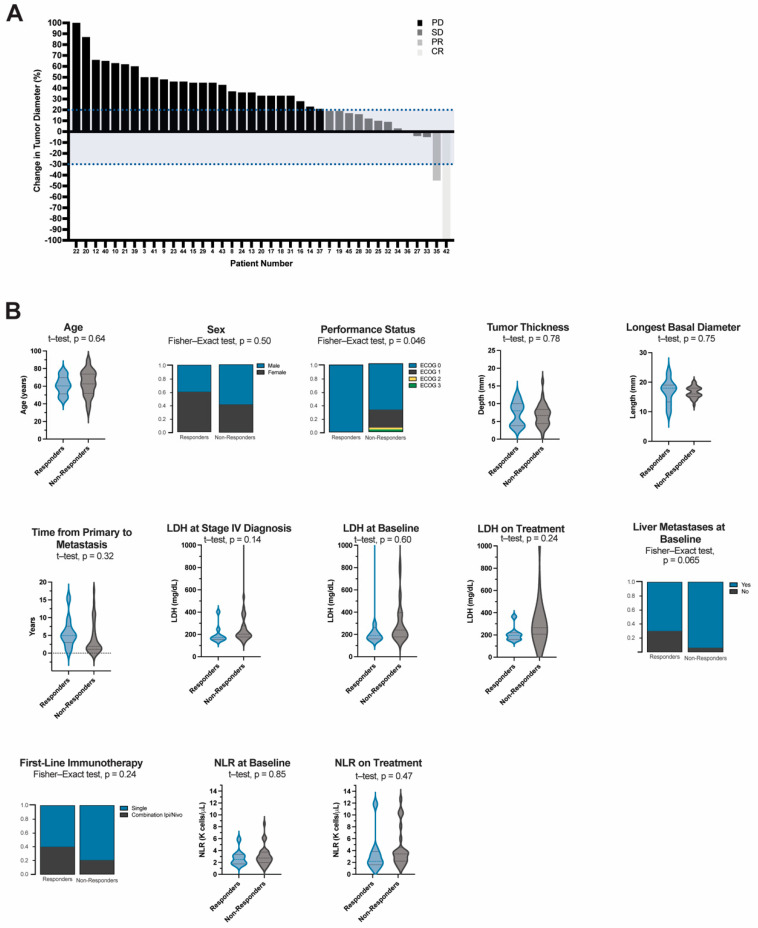
(**A**) Waterfall Plot of Tumor Response using RECIST criteria and (**B**) Demographic, hematologic, and treatment characteristics of responders compared to non-responders. Continuous variables are represented in violin plots with quartiles shown as dotted lines. Independent *t*-test was used to evaluate the difference in mean values between responders and non-responders. Categorical variables are represented by proportions of the analyzed variable within each group. Fisher-exact test was used for group comparisons. Responders (*n* = 10) are comprised of patients who derived a clinical benefit from treatment, i.e., complete response (CR (*n* = 1)), partial response (PR (*n* = 1)), and those with stable disease (SD (*n* = 8)). Non-responders (*n* = 34) are comprised of all those who had progressive disease (PD). Objective response was determined using RECIST criteria. LDH = lactate dehydrogenase, NLR = neutrophil:lymphocyte ratio.

**Table 1 cancers-14-05789-t001:** Patient Characteristics.

Patient Demographics			Univariable Analysis			
			OS		PFS	
Parameter	Categories	Number (%), *n* = 46 (100%)	HR (95% CI)	*p*-Value	HR (95%CI)	*p*-Value
Age (years)	Median (IQR)	61.8 (20.0)	1.0 (0.9, 1.0)	0.33	1.0 (1.00, 1.02)	0.89
Sex	Male	26 (56.5)	0.7 (0.4, 1.4)	0.35	0.8 (0.4, 1.5)	0.46
	Female	20 (43.5)				
ECOG Performance Status	0	33 (71.7)	1.9 (0.9, 3.8)	0.073	1.9 (1.0, 3.7)	0.076
	1	11 (23.9)				
	2	1 (2.2)				
	3	1 (2.2)				
	4	0 (0)				
Sites of Metastasis at Baseline	Liver	41 (89.1)	2.0 (0.6. 6.6)	0.25	1.3 (0.5, 3.6)	0.65
	Lung	21 (47.5)	1.1 (0.6, 2.1)	0.8	0.9 (0.5, 1.7)	0.8
	Brain	3 (6.5)	1.0 (0.3, 3.5)	0.95	0.7 (0.2, 2.4)	0.62
	Bone	11 (23.9)	1.2 (0.6, 2.5)	0.6	1.3 (0.7, 2.6)	0.48
	LN	14 (30.4)	2.1 (1.1, 4.2)	0.031	1.5 (0.8, 2.8)	0.24
	Other	20 (43.5)	0.6 (0.3, 1.2)	0.13	0.7 (0.4, 1.3)	0.26
	Liver Only	14 (30.4)				
	Liver + Extrahepatic. (vs. Liver Only)	28 (60.9)	1.4 (0.6, 3.2)	0.38	0.8 (0.4, 1.5)	0.48
	Extrahepatic Only. (vs. Liver Only)	4 (8.7)	0.5 (0.1, 2.3)	0.36	0.7 (0.2, 2.6)	0.63

**Table 2 cancers-14-05789-t002:** Treatment Characteristics.

Treatment Characteristics			Univariable Analysis			
			OS		PFS	
Parameter	Categories	Number (%), *n* = 46 (100%)	HR (95% CI)	*p*-Value	HR (95%CI)	*p*-Value
Enucleation	No	22 (47.8)				
	Yes	24 (52.2)				
Plaque RT	No	20 (43.5)				
	Yes	26 (56.5)				
Lines of Prior Therapy	0	43 (93.5)	1.3 (0.8, 2.3)	0.30	1.0 (0.6, 1.8)	0.89
	1	2 (4.3)				
	> 1	1 (2.2)				
Immunotherapy	Single ICI	35 (76.1)	2.5 (0.9, 6.6)	0.053	2.0 (0.9, 4.1)	0.076
	Combination ICI	11 (23.9)	0.4 (0.2, 1.0)	0.053	0.5 (0.2, 1.1)	0.076
Cycles of ICI Completed	Median (IQR)	10 (8)	0.9 (0.9, 1.0)	0.009	0.9 (0.9, 1.0)	0.003
Reason for Discontinuation	Progression/Death	31 (67.4)				
	Toxicity	11 (23.9)				
	Other	4 (8.7)				
IRAE Grade	None	24 (52.2)				
	Grade 1	7 (15.2)				
	Grade 2	9 (19.6)				
	Grade 3	6 (13.0)				
	Grade 4	0 (0)				
Liver Directed Therapy	None	23 (50.0)	0.9 (0.5, 1.8)	0.8	1.0 (0.5, 1.7)	0.88
	Surgical Resection	2 (4.3)				
	SBRT/RT	9 (19.6)				
	TACE	6 (13.0)				
	Radioembolization	1 (2.2)				
	Multiple	5 (10.9)				
Objective Response Rate	CR	1 (2.2)				
	PR	1 (2.2)				
	SD	8 (17.4)				
	PD	34 (73.9)				
	Unable to Assess	2 (4.3)				

IRAE, Immune-related adverse event.

**Table 3 cancers-14-05789-t003:** Hematologic Biomarkers.

Hematologic Biomarkers			Univariable Analysis			
			OS		PFS	
Parameter	Categories	Number (%), *n* = 46 (100%)	HR (95% CI)	*p*-Value	HR (95% CI)	*p*-Value
LDH at Stage IV Diagnosis	Median (IQR)	209 (83)				
	WNL (< 240 mg/dL)	29 (63.0)	2.2 (1.1, 4.3)	0.038	1.6 (0.8, 3.2)	0.15
	Elevated (> 240 mg/dL)	14 (30.4)				
	Unknown	3 (6.5)				
LDH at Baseline	Median (IQR)	199 (76)				
	WNL (< 240 mg/dL)	24 (52.2)	3.6 (1.9, 7.0)	0.0011	2.8 (1.5, 5.4)	< 0.0001
	Elevated (> 240 mg/dL)	19 (41.3)				
	Unknown	3 (6.5)				
LDH on Treatment	Median (IQR)	224 (175)				
	WNL (< 240 mg/dL)	18 (39.1)	3.7 (1.6, 8.8)	0.0046	2.2 (1.1, 4.3)	0.0014
	Elevated (> 240 mg/dL)	16 (34.8)				
	Unknown	12 (26.1)				
NLR at Baseline	Median (IQR)	2.7 (1.5)				
	Below Median	21 (45.7)	1.0 (0.5, 2.0)	0.98	1.3 (0.7, 2.5)	0.18
	≥ Median	23 (50.0)				
	Unknown	2 (4.3)				
NLR on Treatment	Median (IQR)	3.2 (1.9)				
	Below Median	16 (34.8)	1.6 (0.8, 3.4)	0.057	1.9 (1.0, 3.9)	0.0098
	≥ Median	19 (41.3)				
	Unknown	11 (23.9)				
dNLR at Baseline	Median (IQR); *n* = 45	1.7 (0.8)				
dNLR on Treatment	Median (IQR); *n* = 36	1.8 (1.0)				
ΔNLR on Treatment	Median (IQR); *n* = 36	0.6 (1.1)				
ALC at Baseline	Median (IQR), *n* = 44	1.5 (0.8)				
	Below Median	22 (47.8)	0.9 (0.5, 1.8)	0.81	0.8 (0.4, 1.4)	0.38
	≥ Median	22 (47.8)				
	Unknown	2 (4.3)				
ALC on Treatment	Median (IQR); *n* = 35	1.7 (1.0)				
	Below Median	17 (37.0)	0.8 (0.4, 1.7)	0.52	0.7 (0.3, 1.4)	0.28
	≥ Median	18 (39.1)				
	Unknown	11 (23.9)				
ANC at Baseline	Median (IQR); *n* = 44	4.1 (1.3)	0.9 (0.5, 1.7)	0.73	1.2 (0.6, 2.2)	0.59
	Below Median	21 (45.7)				
	≥ Median	23 (50.0)				
	Unknown	2 (4.3)				
ANC on Treatment	Median (IQR); *n* = 35	4.9 (3.4)	0.9 (0.4, 1.8)	0.70	0.9 (0.5, 1.9)	0.85
	Below Median	17 (37.0)				
	≥ Median	18 (39.1)				
	Unknown	11 (23.9)				
AEC at Baseline	Median (IQR); *n* = 44	0.1 (0.2)	0.5 (0.2, 1.2)	0.11	0.7 (0.3, 1.8)	0.48
	Below Median	5 (10.9)				
	≥ Median	39 (84.8)				
	Unknown	2 (4.3)				
AEC on Treatment	Median (IQR); *n* = 35	0.2 (0.3)	1.2 (0.6, 2.6)	0.61	0.8 (0.4, 1.6)	0.50
	Below Median	18 (39.1)				
	≥ Median	17 (37.0)				
	Unknown	11 (23.9)				

LDH, lactate dehydrogenase; NLR, neutrophil:lymphocyte ratio; ALC, absolute lymphocyte count; ANC, absolute neutrophil count; AEC, absolute eosinophil count.

## Data Availability

Not applicable.

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
