# Peer review of "Prognostic Hematologic Biomarkers Following Immune Checkpoint Inhibition in Metastatic Uveal Melanoma"

_cancers, 2022, doi:10.3390/cancers14235789_

Round 1

Reviewer 1 Report

Please add the simple summary.

The paper intends to study the usefulness of serological markers for the outcome of ICI treatment. For this, LDH and some blood markers (NLR is NOT a serologic biomarker, it is a blood cell marker)  are analysed in comparison to partial or no responses after ICI. This is a very worthwhile study. However, serologic markers are also prognostic without ICI treatment (see ref Meijer N, Ophthalmology Science June 2022, DOI.org/10.1016/j.xops.2022.100117)

I would strongly advise the authors to focus on their question, and bot bring in other markers such as primary tumor characteristics. You did not investigate the relation between the primary tumor characteristics and LDH or NLR. My strong advise is therefore to delete table 1 till Sites of metastases  and please delete primary tumor data from table 1B.

You can instead add other markers: If you studied other markers than LDH (such as CRP), please add that information. You should also provide prognostic value of the below and above median values of the other blood markers (ALC, ANC, AEC)

What is the use of describing the original intraocular tumor and studying their OS and PFS? You ask about the relation between blood serology and the effect of ICIs, not the time between diagnosis of the primary UM and metastases development/death. Do you have the information on cause of death?

Minor

Abstract: What is the difference in time between at the start of ICI and on treatment? A p value < 0.001 should be changed to p < 0.001, not p=0046, etc.

It is not clear to me what you mean with: a hard cut-off of 60 days following therapy start was used for all on treatment assessments. What moment did you analyse?? How many did you exclude then?

Methods; WBC: Please add the full terminology

In M and M, mention how many had CR, PR, SD. According to M and M, you use the term PR both for partial response and progressive disease. Please separate.

For OS and PFS, you use time to death from any cause. Why not death from metastatic disease?

Reviewer 2 Report

The manuscript “Serologic biomarkers of response to immune checkpoint inhibitors in metastatic uveal melanoma” addresses the interesting issue of markers of response to therapy with immune checkpoint blockers (ICB). The study recognizes the small patient number as a limitation of the study.

The tumor mutational burden (TMB) is known to be very low for uveal melanoma but high in cases with MBD4 mutation. The authors should analyze TMB or at least MBD4 mutations since the few cases showing a response to ICB might carry this mutation or otherwise show high TMB and the serological biomarkers might incidentally correlate with these features.

Minor remarks

Patients were treated between 2006 and 2022 receiving various types of ICB. Ipilimumab was the first ICB to be approved for melanoma in 2011. It is therefore not clear which procedure was followed to treat the patients between 2006 and 2011.

“Less than 5% of patients were lost to follow up.” The exact number should be given.

Reviewer 3 Report

The manuscript titled: “Serologic biomarkers of response to immune checkpoint inhibitors in metastatic uveal melanoma” is well written. The manuscript is based on a well-constructed scientific concept and carried out the studies are well. However, data needs to be refined in a presentable manner. The present manuscript would be benefited by addressing the points below. I would suggest accepting the manuscript after minor revision. 

Comments:

·      Authors should write the simple summary as journal guidelines, which is missing from the manuscript.

·      Adjust the size of tables in the manuscript as per the journal guidelines.

Round 2

Reviewer 1 Report

This is a nice new approach to try to identify patients with a response to ICI's.

Author Response

Thank you for your comments. We appreciate your thorough review. 

Reviewer 2 Report

If the samples in the diagnostic workout have not been analyzed by sequencing this should be done now. The conclusion that specific serum biomarkers indicate response to immunotherapy could be false if the MBD4 mutation and serum marker expression would incidentally correlate. Hence, the conclusion that the markers identified predict response to immunotherapy cannot be drawn. 
